# Potential Ecological Distribution of the Beetle *Agrilus mali* Matsumura (Coleoptera: Buprestidae) in China under Three Climate Change Scenarios, with Consequences for Commercial and Wild Apple Forests

**DOI:** 10.3390/biology13100803

**Published:** 2024-10-08

**Authors:** Yanlong Zhang, Hua Yang, Aerguli Jiamahate, Honglan Yang, Liangming Cao, Yingqiao Dang, Zhaozhi Lu, Zhongqi Yang, Tohir A. Bozorov, Xiaoyi Wang

**Affiliations:** 1Key Laboratory of Forest Protection of National Forestry and Grassland Administration, Ecology and Nature Conservation Institute, Chinese Academy of Forestry, Beijing 100091, China; zhangyl@caf.ac.cn (Y.Z.); caolm1206@126.com (L.C.); yqdang@caf.ac.cn (Y.D.); yangzhqi@126.com (Z.Y.); 2College of Forestry, Sichuan Agricultural University, Chengdu 611130, China; yanghua151017@163.com; 3State Key Laboratory of Desert and Oasis Ecology, Key Laboratory of Ecological Safety and Sustainable Development in Arid Lands, Xinjiang Institute of Ecology and Geography, Chinese Academy of Sciences, Urumqi 830011, China; 18040921335@163.com (A.J.); yanghonglan@ms.xjb.ac.cn (H.Y.); tohirbozorov@yahoo.com (T.A.B.); 4College of Plant Health and Medicine, Qingdao Agricultural University, Qingdao 266109, China; zhaozhi_lv@sina.com; 5Laboratory of Molecular and Biochemical Genetics, Institute of Genetics and Plants Experimental Biology, Uzbek Academy of Sciences, Yukori-Yuz, Kibray 111226, Tashkent Region, Uzbekistan

**Keywords:** apple jewel beetle, climate change, expansion of distribution, MaxEnt model, wild apple trees

## Abstract

**Simple Summary:**

The apple buprestid beetle *Agrilus mali* is a pest that can cause serious harm to various domestic apples and many species of *Malus*. Currently, it is distributed in 17 provinces of China. After spreading from Shandong province to Xinjiang Uygur Autonomous Region, it has caused great damage to the local wild apple resources. China is a major producer of apples; once this pest spreads, it will seriously affect China’s apple industry. We used the MaxEnt model to analyze the changes in the distribution area of *A. mali* under future climate change scenarios in China. The forecasted suitable regions for *A. mali* in China will expand northward in the 2050s and 2070s. The forecasted highly suitable regions were 1.11–1.34 times larger than they are currently, and their central distributions were 61.57–167.59 km further north. These findings suggest that monitoring and management measures should be implemented urgently to protect the commercial apple industry and wild apple resources alike.

**Abstract:**

The apple jewel beetle (AJB), *Agrilus mali* Matsumura (Coleoptera: Buprestidae), is a dangerous pest of commercial apple orchards across China, the largest apple production country in the world, and has recently become invasive in the Xinjiang Uygur Autonomous Region (XUAR) of northwestern China, where wild apple forests also occur. This pest poses a serious threat to apple production and wild apple forests throughout the world. Global warming is expected to change the geographical distribution of *A. mali* in China, but the extent of this is unknown. Based on empirical data from 1951 to 2000, a MaxEnt model was used to forecast the ecological distribution of *A. mali* under three different climate scenarios projected in the fifth report of the Intergovernmental Panel on Climate Change. The results showed that the most important variables were the maximum temperature of November, precipitation in January, and minimum temperatures in April. Under all climate scenarios, the forecasted suitable regions for *A. mali* in China will expand northward in the 2050s and 2070s. The forecasted highly suitable regions will be 1.11–1.34 times larger than they are currently, and their central distributions will be 61.57–167.59 km further north. These findings suggest that the range and damage caused by *A. mali* in China will increase dramatically in the future. Monitoring and management measures should be implemented urgently to protect both the commercial apple industry and wild apple resources.

## 1. Introduction

The apple jewel beetle *Agrilus mali* Matsumura (Coleoptera: Buprestidae), an important pest of apple trees, is distributed across the Russian Federation, the Republic of Mongolia, the Korean Peninsula, Japan, and 17 provinces and regions in China. In 1993, *A. mali* successfully established itself in the Xinjiang Uyghur Autonomous Region, in northwestern China [1,2], with disastrous consequences for the wild apple forests (*Malus sieversii* (Ledeb.) Roem) and cultivated apple trees to the north and south of the Tianshan Mountains [2,3,4,5,6,7]. Adult beetles feed on apple leaves with little damage to the trees; however, the larvae feed on the bark, phloem, and xylem, causing severe damage [3,8]. By 2002, *A. mali* had infested 35% of wild apple trees in Gongliu County, Xinjiang [9], and, subsequently, killed tens of thousands of wild apple trees and infested over 80% of the total area of wild apple forests in the Yili River Valley in Xinjiang [4]. This particular wild apple species, *Malus sieversii*, is only found in the mountains of Central Asia, including southern Kazakhstan, eastern Uzbekistan, Kyrgyzstan, Tajikistan, Turkmenistan, and China (Xinjiang) [10,11]. It is considered the key ancestor of cultivated apples [12]. These wild apple trees exist in patches, and the distance between patches is not far, so *A. mali* could easily cross barriers and spread from Xinjiang to wild fruit forests in neighboring countries [13].

In addition to Xinjiang, within China, *A. mali* is also found in the Heilongjiang, Jilin, Liaoning, Hebei, Henan, Shandong, Shanxi, Shaanxi, Gansu, Ningxia, Hubei, Qinghai, Jiangsu, and Sichuan provinces, as well as in Beijing and in the Inner Mongolia Autonomous Region [2,4,5,14]. Large areas of cultivated apple trees (*Malus domestica*) are grown in these regions, where *A. mali* has caused huge losses [15,16], and it has the potential to threaten other apple-growing areas in China. In 2016, China produced approximately 44 million tons of apples from a planting area of 2.2 million hectares, accounting for 49% and 44% of the world’s apple production and cultivated area, respectively [17]. It is vital to prevent both cultivated and wild apple trees from being further damaged by *A. mali*.

Insects are poikilothermal organisms whose body temperatures fluctuate in relation to variations in environmental temperatures, and the metabolism rates of key physiological processes, driven by ambient temperatures, determine their population dynamics and distribution [18,19]. The Intergovernmental Panel on Climate Change (IPCC) predicted that the average global surface temperature from 1990 to 2100 would rise by values between 1.4 °C and 5.8 °C [20]. These increased temperatures will have a significant impact on poikilothermal animals such as forest insects [21], accelerating their life cycle and shortening the developmental time of each generation. We already know that differences in temperature between the regions where *A. mali* is distributed have led to differences in its life history: in the northernmost part of China, *A. mali* requires two years to complete a generation, while in the Hubei and Jiangsu provinces further south, only one year is required [2]. In addition, adult *A. mali* specimens appear one month earlier in the lower latitude provinces of Hubei and Jiangsu than in the higher latitude provinces of Heilongjiang and Xinjiang [2]. Because of climate warming, the geographical ranges of insect species like *A. mali* may expand northward in the northern hemisphere and, simultaneously, to higher altitudes [22,23].

The maximum entropy model (MaxEnt) has been widely used to predict the distribution of animals, plants, and algae [24,25,26]. The parameters of selected environmental variables in locations where the model organism is or has been present are used to forecast the distribution of the organism under changed environmental conditions [27]. For example, the potential habitat and the current and future distribution of the invasive insect pest *Ricania shantungensis* Chou and Lu (Hemiptera: Ricaniidae) in Korea were predicted using a MaxEnt model [28]. The distribution range across North America of emerald ash borer *Agrilus planipennis* Fairmaire (Coleoptera: Buprestidae), which belongs to the same genus as *A. mali*, was also predicted using a MaxEnt model [27].

There have been few studies on the spread of *A. mali* in China. A risk analysis in the Shaanxi province showed that *A. mali* was a highly dangerous pest with a risk (R) value of 2.42 [29]. The spatial distribution of larvae in wild fruit forests has shown an aggregated distribution pattern [30]. Up until now, there has been no report about the macro-spatial distribution pattern and dominant climatic and environmental factors affecting the geographical distribution of *A. mali* in China.

To explore the potential distribution area and development of *A. mali* in China under a backdrop of global warming, it was first necessary to establish comprehensively the current distribution of *A. mali* in China and measure environmental variables where this pest is present. Based on the geographical distribution data, we then used a niche MaxEnt model combined with GIS to forecast the potentially suitable areas for *A. mali* in China under three future climate scenarios projected in the fifth report of the IPCC. In addition, the main climatic and environmental factors affecting its distribution were analyzed to delimit the geographical distribution and potential risk of *A. mali* spreading in China. In China, a large number of apple growers, planting enterprises, and seedling cultivation enterprises may conduct early monitoring based on our data. Once discovered, measures can be taken as soon as possible to avoid causing greater losses. The yield of apples saved from pests can provide people with more food and nutritional sources, which is in line with the second goal of the United Nations Sustainable Development Goal—eradicating hunger.

## 2. Materials and Methods

### 2.1. Historic and Current Locations of A. mali

The longitude and latitude data of locations where *A. mali* (AJB) was present were recorded by field surveys with a GPS (accurate to two decimal places). Species distribution data were obtained by searching the published literature and the databases of the Southwest China Animal Resources Database (http://www.swanimal.csdb.cn, accessed on 15 June 2024), Global Biodiversity Information Facility (GBIF, http://www.gbif.org/, accessed on 12 June 2024), Center for Agriculture and Biosciences International (CABI, http://www.cabi.org/, accessed on 13 June 2024), and the teaching specimen resource-sharing platform (http://mnh.scu.edu.cn, accessed on 13 June 2024) [31]. Accordingly, 194 locations where *A. mali* was present or recorded were obtained. Repetitive, fuzzy, and adjacent records were deleted as per the requirements of MaxEnt [32]. ENMTOOLs v1.4.1 was used to calculate the distance between each grid center (30 arc seconds) and each set of distribution data; then, the distribution data closest to the center of each grid were retained. All the 128 retained records were imported into the Microsoft Excel software (Microsoft Office 2010) and saved in a CSV format.

### 2.2. Environmental Variables

Twenty-three bioclimatic variables (Table 1) reflecting temperature, precipitation, and seasonal changes were selected [33]. Data from the interval of 1950–2000 were downloaded from the WorldClim website (www.worldclim.org, accessed on 14 June 2024), issued within the fifth evaluation report of the IPCC. Three representative CO_2_ concentration pathways (RCPs) used by the IPCC were added as future climate change scenarios: RCP2.6, RCP4.5, and RCP8.5 [34]. RCP2.6 represents minimum greenhouse gas emissions, and RCP8.5 represents the maximum. We compared the two medium-emission scenarios, RCP4.5 and RCP6.0, and selected RCP4.5 to use in our MaxEnt model [35]. The future periods to model were the 2050s (2041–2060) and 2070s (2061–2080), and data were downloaded from the International Centre for Tropical Agriculture (CIAT). The spatial resolution of the data was 30 arc seconds (1 km).

Using Worthington’s method [36], 13 environmental variables (Table 1) were selected for analysis to determine their contribution to the climate suitability model for the current and future distribution of *A. mali* in China. The permutation importance was measured until the final performance of the model rather than the path used in an individual run, and, therefore, it was better to evaluate the importance of a particular variable [37].

### 2.3. Modeling Methods and Statistical Analysis

The MaxEnt model (version 3.4.1) used in this study is freely available at http://www.cs.princeton.edu/ (accessed on 19 June 2024). Data associated with the 128 *A. mali* distributed locations and their environmental variables were imported into the software. The Kuenm R package (v1.1.10) was used to optimize the regularization multiplier (RM) (0.5, 1, 1.5, 2, 2.5, 3, 3.5, 4) and feature combination (FC, including 30 types) of the model, and the optimal setting of the minimum information criterion AICc value (delta.AICc) among 240 results was selected. Seventy five percent of the *A. mali* distribution data locations was randomly selected as the training data to establish the prediction model, and the remaining twenty five percent was used to test the model. A Jackknife test was used to measure the weight of each variable to create an environmental variable response curve. The default values of the model were applied to other parameters [38]. The area under the receiver operating characteristic (ROC) curve is the AUC value (area under curve). The AUC is not affected by prevalence or diagnostic thresholds, which can jointly compare the accuracy of the two diagnostic tests. Larger average AUC values indicate a better prediction accuracy for the model. AUC values can range between 0.5 and 1.0 (Table 2) and are divided into five classes [39]. The closer the AUC for a model comes to 1, the better is the model’s performance.

The MaxEnt model estimates the probability that a species will occur in a particular environment based on known occurrence records and randomly generates background points by finding the maximum entropy distribution. A distribution model was obtained based on the average logistic outputs of 10 replicated runs used to estimate the probability of occurrence between 0 (not likely to occur) and 1 (most likely to occur) [40]. Model predictions were imported into a GIS format for generating maps using ArcMap 10.0. The suitability of a region for an *A. mali* occurrence was categorized as follows: existence probability <0.05, unsuitable region; 0.05≤ − <0.33, poorly suitable region; 0.33≤ − <0.66, moderately suitable region; and ≥0.66, highly suitable region.

### 2.4. Calculation of Geometric Center and Displacement

Referring to the calculation methods of Yue et al. [41], the variation in area and the mean center of the total and highly suitable distribution in different periods and different times were calculated as follows: $$\left\{\begin{array}{c} x\left(t\right)=\sum_{i=1}^{I}\frac{Si\left(t\right)*Xi\left(t\right)}{S\left(t\right)}\\ y\left(t\right)=\sum_{i=1}^{I}\frac{Si\left(t\right)*Yi\left(t\right)}{S\left(t\right)} \end{array}\right.$$
where *t* is the time period, *I* is the number of the unit grid of the total or high-suitability zone, *Si*(*t*) is the area of the unit grid of the *t* period, and *S*(*t*) of the *t* period is the total area of the total or high-suitability zone of the *t* period. *Xi*(*t*), *Yi*(*t*) is the centroid coordinate of the unit grid of the total or high-suitability zone in the *t* period, and *x*(*t*), *y*(*t*) is the centroid coordinate of the total or high-suitability zone in the *t* period.
$$D=\sqrt{{\left(x\left(t+1\right)-x\left(t\right)\right)}^{2}+{\left(y\left(t+1\right)-y\left(t\right)\right)}^{2}}$$
$$\theta =arctg\left(\frac{y\left(t+1\right)-y\left(t\right)}{x\left(t+1\right)-x\left(t\right)}\right)$$
where *D* is the displacement distance from *t* period to *t* + 1 period of the total or high-suitability region; *θ* is the direction of displacement of the total or high-suitability region from *t* period to *t* + 1 period; 0° < *θ* < 90° means that the direction of displacement was northeast; 90° < *θ* < 180° means that the direction of displacement was northwest; 180° < *θ* < 270° means that the direction of displacement was southwest; and 270° < *θ* < 360° means that the direction of displacement was southeast.

## 3. Results

### 3.1. Evaluation of the MaxEnt Model

The initial environmental variables selected in this study included 19 bioclimatic variables and monthly average climate data commonly used to predict species distribution (Table 1). The AUC value of the training data was 0.992 (Figure 1), which was very close to 1.0 and suggested that the model described the data at an excellent level. The Jackknife test showed that the maximum temperature of November (tmax11), precipitation in January (prec1), and minimum temperatures of April (tmin4) contributed more to the model than other variables (Figure 2 and Figure 3). Following these three variables in terms of importance were the precipitation in December (prec12), the mean temperature in November (tmean11), and the maximum temperature in February (tmax2) (Figure 3).

### 3.2. Spatial Pattern of A. mali Distribution under Global Warming

Currently, the total suitable region for the occurrence of *A. mali* in China is between 25.23°–49.66° N and 78.98°–134.58° E (Figure 4). The highly suitable regions are northwest of Xinjiang, south of Gansu, Ningxia, Shaanxi, south of Shanxi, Heibei, Beijing, Tianjin, Liaoning, Jilin, south of Heilongjiang, north of Henan, Shandong, northeast of Sichuan, and east of Inner Mongolia (Figure 4). Under the three climate change scenarios, the highly suitable regions for *A. mali* in China expanded significantly, especially in the 2050s under scenarios RCP2.6 and 2070s under scenario RCP2.6 (Figure 5). Under these scenarios, the Shaanxi and Shanxi provinces would become highly suitable regions, and the Yunnan province and Chongqing city would also become a suitable region for *A. mali* for the first time.

The analysis of changes in the centroid of the total suitable regions for *A. mali* showed that it shifted northwards under the 2050s and 2070s climate change scenarios compared to the current total suitable region (Table 3). The meridional moving distance was 23.97–56.65 km, and the angle was 42.84–204.74° (Table 3). Under the same climate change scenario, the centroid of the total suitable distribution area for *A. mali* in the 2070s would be further north than that in the 2050s (Table 3). The analysis of the centroid of the highly suitable region for *A. mali* occurrence shows that, under these scenarios, it would move north by 61.57–167.59 km and the angle would be 23–41.38° in the 2050s and the 2070s compared to the centroid of the current highly suitable region (Table 4). 

### 3.3. Change in the Distribution Area of A. mali under Global Warming Scenarios

The forecasted suitable region for *A. mali* in China increased under all climate change scenarios for both time periods (Figure 6). The MaxEnt model determined that the suitable region for *A. mali* was 26,027,645 km^2^, and the highly suitable region was 1,091,466 km^2^, among which the Beijing and Tianjin locations were the most dangerous, accounting for 100% of the whole area. The ranges of suitable regions revealed almost no changes under different climate scenarios in 2050s and 2070s, but the area change in a highly suitable region was obvious. The maximum and minimum areas covered by a highly suitable region would be 1,457,640 km^2^ and 1,216,426 km^2^, respectively, in the scenarios of 2050s RCP2.6 and 2070s RCP8.5, that is, 1.34 and 1.11 times higher than the current area (Figure 6). 

## 4. Discussion

The model results showed that the current suitable region for *A. mali* was 25.23°–49.66° N and 78.98°–134.58° E, mainly in the north, northeast, and northwest of China. Under global warming, the highly suitable area is forecasted to expand, and the centroid of this area shows a northward migration. This result is similar to previous studies showing that, as the temperature increases, the geographical ranges of insect species may simultaneously shift northward in latitude and upwards in altitude [22,23,42]. For example, the winter pine processionary moth (*Thaumetopoea pityocampa*) underwent an extraordinary expansion to high-elevation pine stands in the Italian Alps; its altitudinal range limit increased by one third of the total altitudinal expansion over the last three decades [43]. According to historical outbreak records, the distributions of both winter moth (*Operophtera brumata*) and autumn moth (*Epirrita autumnata*) have expanded northward in Scandinavia [44]. The rise in the lowest temperature in winter led to the outbreak of the southern pine beetle (*Dendroctonus frontalis*) in North America [45]. Our modeling shows that some areas, such as Chongqing, will become infested by *A. mali* for the first time. The increase in the infestation degree and the expansion of suitable regions will increase the difficulties in controlling this pest. 

Modeling showed that the highly suitable regions for *A. mali* occurrence in China expanded the most in the 2050s under the climate change scenario of RCP2.6 and in the 2070s under the climate change scenario of RCP2.6. The increase in the highly suitable area for *A. mali* in the Shaanxi and Shanxi provinces is particularly concerning, as Shaanxi is the largest apple cultivation province in China. Its output in 2018 accounted for about a quarter of China’s production and one-sixth of the world’s production [46]. Such a concentrated host tree distribution provides excellent conditions for the development of large *A. mali* populations. Therefore, an early warning system should be established to monitor the emergence of adults, and a timely warning should be issued to all apple production areas to avoid outbreaks. In addition, it is also necessary to prevent the movement of wood materials among the different apple-growing areas. For forest insects, trade in live plants and the transport of wood packaging materials and firewood are considered the most important pathways facilitating long-distance invasions [47,48]. Apple trees need to be pruned every year, which produces many branches and trunks potentially infested with *A. mali* larvae. The transport of such material to an area where *A. mali* is not present is very dangerous and not allowed in China. 

The potential risk of *A. mali* spreading to other apple production areas cannot be ignored. Besides the Shaanxi and Shanxi provinces, apple trees are planted in 23 other provinces of China, but *A. mali* has not been found in 9 of them. Once a new area is infested by this pest, an outbreak may occur rapidly under the appropriate climate conditions. In 1991, *A. mali* was found in apple orchards in Pingliang City, in the Gansu province, with an infestation rate of 86%. Many apple orchards were completely destroyed within five years [15]. In the late 1990s, *A. mali* caused serious damage to the apple orchards in Jianzha County, Qinghai province. Around 45% of the orchards were infested, and 60% of the infested orchards were seriously damaged, killing 20% of the trees [16]. Under future climate change scenarios, the area highly suitable for *A. mali* could encompass the most suitable planting areas for apples in China [17]. Thus, different control measures should be taken in different areas: for areas without *A. mali*, the introduction of apple plant materials such as seedlings, scions, and wood materials from other areas should be limited; and, for areas with records of this pest, we should strengthen monitoring and prediction of their populations.

Our study showed that there was redundancy between some of the initial environmental variables selected for the MaxEnt model, which affected the prediction results, similar to previous research [49,50]. In order to establish a high-precision model, the Jackknife test was used to evaluate each variable’s contribution (≥5%), and the importance of permutations (≥5%) to the simulation from the initial 23 environmental variables by using Worthington’s method [36]. Six of the most relevant variables were selected; besides the variables of prec1 and prec12 (precipitation in January and December), all other variables were related to temperature.

Bioclimatic environmental factors, especially temperature and rainfall, would affect not only the physiological activities of insects directly but also indirectly influence their population and distribution by limiting their food sources and habitat conditions [51]. This is confirmed by the dominant climate factors (tmax11, prec1, tmin4, prec12, tmean11, and tmax2), selected according to the Jackknife test in the model. According to the literature and our observations, *A. mali* specimens usually overwinter as larvae from November to April [2]. So, the temperatures in November, March, and April are important factors, affecting the beginning and end of overwintering. Many insects are forced to diapause in order to survive the winter [52]. The diapause behavior of wood-borers is affected by temperature much more than by the photoperiod. Temperature influenced the voltinism of the bark beetles *Ips typographus* and *Dendroctonus rufipennis* through its effect on diapause [53], and *Osmia lignaria* required a period of cold temperature for winter diapause completion [54]. Like the species above, the highest and average temperature in November (tmax11 and tmean11) determined whether *A. mali* could experience a low enough temperature to enter diapause. Temperatures in early spring play an important role in diapause termination under natural conditions. For example, for the beet webworm *Loxostege sticticalis*, the threshold temperature for post-diapause development in pupae was 10.60 °C, with a corresponding accumulation of 200 degree days [55]. The mild winter temperatures may be detrimental to some overwintering insects [56], and the temperature change (tmax2) in February will affect the diapausing larvae. In April, the larvae begin to feed, and low temperatures are not conducive to the development of larvae. In most apple-growing areas of China, precipitation in December and January is snow, which is unlikely to affect the overwintering larvae under the bark directly. We speculate that precipitation in the winter could affect hypodermic larvae through the host.

The predicted suitable region for the occurrence of a species is the result of the integrated effects of various factors, but, theoretically, only abiotic factors need to be considered [57]. However, in the actual living environment, human factors and some biotic factors (the interactions among species, vegetation types, and the dispersal capacity of species) will also impact the potential distribution of species. As a wood-borer, *A. mali* is directly affected by external climatic conditions when it is outside of the tree (adult period). During its immature stages inside trees (most of its life cycle), this pest may be slightly influenced by the climate. In addition, the distribution of wood-borers is also affected by susceptible host plants [58]. If the host factors were to be put into the MaxEnt model as environmental variables, more accurate results could be obtained. However, the distribution data of susceptible hosts are insufficient for analysis at present and need to be further studied in the future.

## 5. Conclusions

This study used the historical and current locations of *A. mali* in its native range to forecast its distribution in the 2050s and 2070s in China under three climate change scenarios (RCP2.6, RCP4.5, and RCP8.5). The forecasted distribution area of *A. mali* expanded to the north under all scenarios. The future expansion of *A. mali* to new areas is a major threat to both cultivated apple and wild apple forests in China and around the world.

## Figures and Tables

**Figure 1 biology-13-00803-f001:**
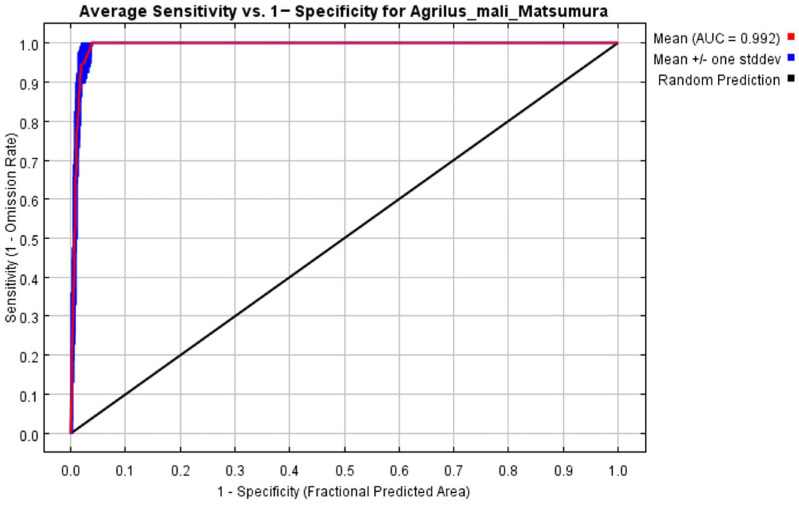
Receiver operating characteristic (ROC) curve and area under the ROC (AUC) values in the period 1950–2000.

**Figure 2 biology-13-00803-f002:**
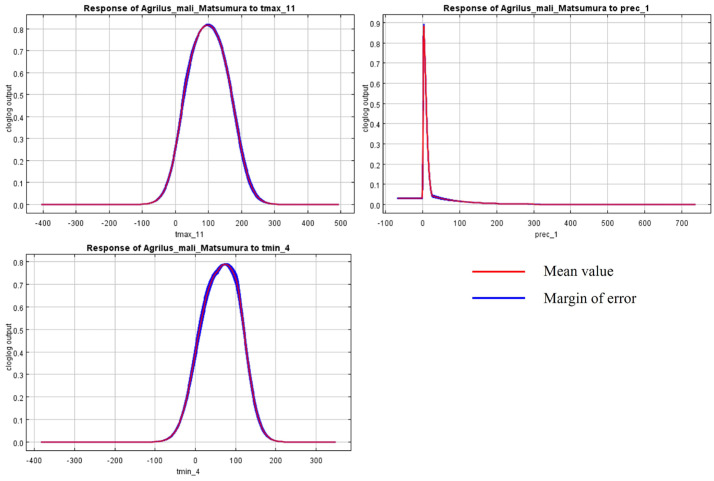
Response curves of *A. mali* in relation to the three most important environmental variables.

**Figure 3 biology-13-00803-f003:**
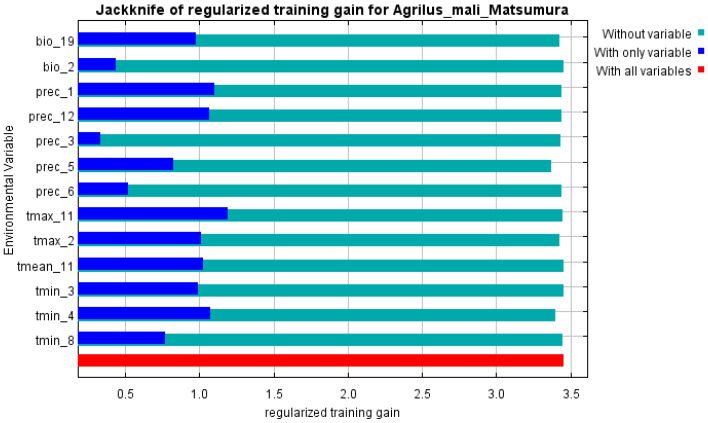
Jackknife test of variables’ importance for the *A. mali* MaxEnt model.

**Figure 4 biology-13-00803-f004:**
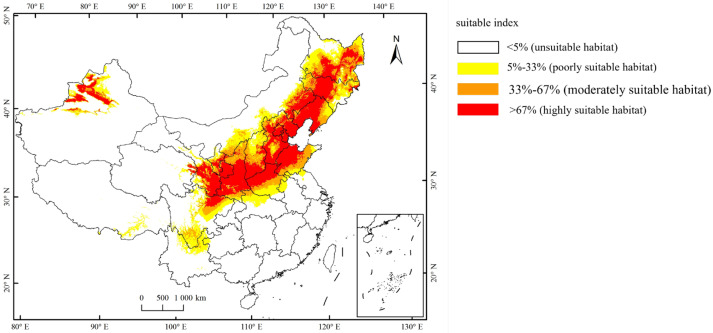
Current suitable climatic regions for *A. mali* occurrence in China.

**Figure 5 biology-13-00803-f005:**
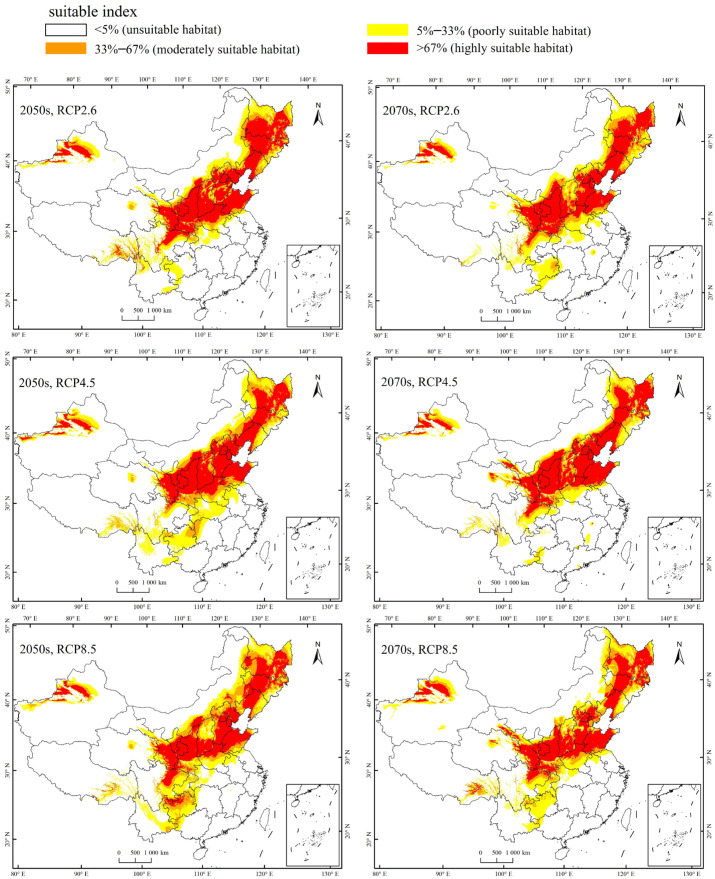
Forecasted suitable climatic regions for *A. mali* occurrence in China in the 2050s and 2070s under three climate change scenarios.

**Figure 6 biology-13-00803-f006:**
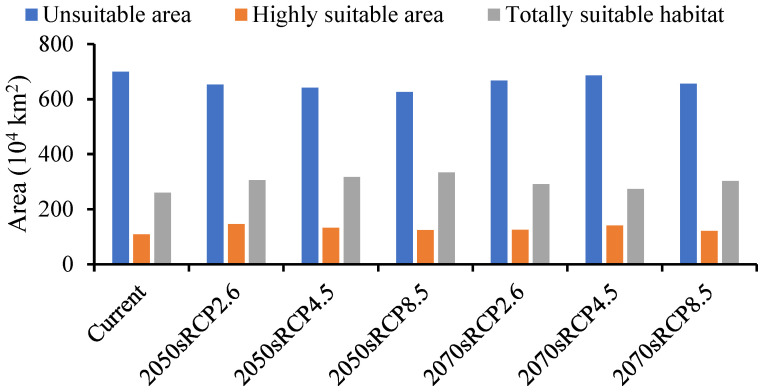
Forecasted area of *A. mali* occurrence in China in the 2050s and 2070s under three climate change scenarios.

**Table 1 biology-13-00803-t001:** List of bioclimatic variables.

Bioclimatic Characteristic	Abbreviation	Time	Abbreviation
Annual mean temperature	bio1		
Mean diurnal range (mean max. temp–mean min. temp)	bio2	Monthly	bio2 *
Isothermality (bio2/bio7) (×100)	bio3		
Temperature seasonality (standard deviation ×100)	bio4		
Max. temperature of warmest month	bio5		
Min. temperature of coldest month	bio6		
Temperature annual range (bio5-bio6)	bio7		
Mean temperature of wettest quarter	bio8		
Mean temperature of driest quarter	bio9		
Mean temperature of warmest quarter	bio10		
Mean temperature of coldest quarter	bio11		
Annual precipitation	bio12		
Precipitation of wettest month	bio13		
Precipitation of driest month	bio14		
Precipitation seasonality (coefficient of variation)	bio15		
Precipitation of wettest quarter	bio16		
Precipitation of driest quarter	bio17		
Precipitation of warmest quarter	bio18		
Precipitation of coldest quarter	bio19	Yearly	bio19 *
Minimum temperature	tmin	March, April, and August	tmin3 *, tmin4 * tmin8 *
Maximum temperature	tmax	February and November	tamx2 *, tamx11 *
Average temperature	tavg		
Precipitation	prec	January, March, May, June, and December	prec1 *, prec3 *, prec5 *, prec6 * prec12 *
Mean temperature of month	tmean	November	tmean11 *
Total number of variables	23		13

Note: * the environmental variables selected.

**Table 2 biology-13-00803-t002:** Model evaluation criterion of the AUC.

AUC Value	Evaluation Criterion (to Describe Reality)
<0.5	Fails
0.5 ≤ AUC < 0.6	Fails
0.6 ≤ AUC < 0.7	Poor
0.7 ≤ AUC < 0.8	Moderate
0.8 ≤ AUC < 0.9	Good
≥0.9	Excellent

**Table 3 biology-13-00803-t003:** Distance and direction shift in the mean center of the total suitable area for *A. mali* under various climate change scenarios.

Period	RCP2.6 SRES-RCP2.6			RCP4.5 SRES-RCP4.5			RCP8.5 SRES-RCP8.5		
	Displacement (km)	Direction	Angle (°)	Displacement (km)	Direction	Angle (°)	Displacement (km)	Direction	Angle (°)
From the present until the 2050s	27.19	Northwest	156.31	37.49	Northwest	160.50	33.64	Southwest	204.74
From the 2050s to the 2070s	42.81	Northeast	7.20	24.91	Northeast	37.11	38.90	Northwest	127.20
From the present until the 2070s	23.97	Northeast	42.84	31.59	Northwest	119.32	56.65	Northwest	162.63

**Table 4 biology-13-00803-t004:** Distance and direction shift in the mean center of highly suitable areas for *A. mali* under various climate change scenarios.

Period	RCP2.6 SRES-RCP2.6			RCP4.5 SRES-RCP4.5			RCP8.5 SRES-RCP8.5		
	Displacement (km)	Direction	Angle (°)	Displacement (km)	Direction	Angle (°)	Displacement (km)	Direction	Angle (°)
From the present until the 2050s	112.44	Northeast	23.00	125.21	Northeast	34.68	68.68	Northeast	23.78
From the 2050s to the 2070s	33.14	Northeast	43.40	46.67	Northeast	5.77	21.13	Northwest	141.99
From the present until the 2070s	143.96	Northeast	27.60	167.59	Northeast	26.95	61.57	Northeast	41.38

## Data Availability

The authors confirm that the data supporting the findings of this study are available within the article.

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
