# Peer review of "Potential Ecological Distribution of the Beetle Agrilus mali Matsumura (Coleoptera: Buprestidae) in China under Three Climate Change Scenarios, with Consequences for Commercial and Wild Apple Forests"

_biology, 2024, doi:10.3390/biology13100803_

Round 1
Reviewer 1 Report
Comments and Suggestions for Authors
The present article addresses an interesting topic related to the distribution of a insect pest of apple plantations in China. The possible influences of climatic changes are considered in the prediction of suitable areas (in the future).
Some specific corrections are necessary:
- on line 33: than currently;
- which is an optimum temperature for the development of Agrilus mali?
- line 91: Emerald ash borer;
- line 92: 2,42 is high or low value?
- lines 108-110: excerpt that antecipates results and discussions?
- line 100: "A. mali" in italics;
- line 120: "adjacent records": how close?
- line 155: better is the model;
- lines 163-165: at the discretion of the authors or should some reference be cited?
- lines 188-189: repetition of line 137?
- pay attention to spaces between characters in lines 205, 209 and 220;
- Figure 8 citeb before Figure 7 in the text; why?
- lines 203 and 244: "mali" not in italics;
- line 254: from largest to smallest values?
- line 322: ...temperatures are the... factors... that affect;
- the reference of Lu et al. 2022 (Bulletin of Entomological Research) could be cited in the discussion.
Author Response
Dear reviewer, thank you very much for your careful reviews of our manuscript. Your suggestions have been of great significance in improving my manuscript. Below, I will respond to each of your suggestions one by one.
- - on line 33: than currently.
Reply: As an adjective, 'current' is incorrect to use here. I accept your suggestion. “Currently” is a noun, so this place should use a noun.
- - which is an optimum temperature for the development of Agrilus mali?
Reply: Agrilus mali is a fully metamorphosed insect that goes through four stages: egg, larva, pupae, and adult. Currently, there is no literature on the optimal temperature for the egg, larva, and pupae stages of A. mali. Only Ma(2020) (master's thesis) had studied the impact of temperature on the population dynamics of adult insects. He counted the number of adult insects trapped in three locations and calculated the average daily temperature. When the average temperature in location 1 was 21.5 ℃, the population of adult insects reached its peak at 2.68 per trap; Location 2 has a peak of 3.12 per trap and a temperature of 22.3℃; The peak value of location 3 is 4.59 per trap, and the daily average temperature is 17.9 ℃. It can be inferred that the temperature range of 17.9 ℃ to 22.3 ℃ is favorable for the activity of adult insects. Cui(2016) recorded that Agrilus mali can emerge at room temperature, indicating that its larvae and pupae can develop at a temperature of 25 ℃ (Cui et al., 2016).Li (2017)studied the lifespan of adult Agrilus mali at room temperature and the feeding of larvae at 25 ℃ (Li et al.,2017). However, none of these documents have studied the optimum developmental temperature for it.
- - line 91: Emerald ash borer.
Reply:Accept,Delete 'the'.
- - line 92: 2.42 is high or low value?
Reply:According to the danger level of harmful organisms in Chinese agriculture, they are classified into four levels. The first level is low risk, 1.0 ≤ R < 1.5; The second level is moderate risk, 1.5 ≤ R < 2.0; The third level is highly dangerous, 2.0 ≤ R < 2.5; The fourth level is particularly dangerous, 2.5 ≤ R < 3.0. So, 2.42 is a high value.
- - lines 108-110: excerpt that antecipates results and discussions?
Reply:This sentence does seem like a result and discussion, what I want to express is the possible situations that may arise through this paper, which can also be considered as the possible significance. I also strongly agree with the expert's opinion to delete it.
- - line 100: "A. mali" in italics.
Reply:Accept. Have made "A. mali" in italics.
- - line 120: "adjacent records": how close?
Reply:We use a 5Km * 5Km grid to filter distribution sites. If two sites fall into one grid, only one distribution point should be retained.
- - line 155: better is the model.
Reply: Accept. Change Line 155 to “The closer the AUC for a model comes to 1, means the better is the model.”
- - lines 163-165: at the discretion of the authors or should some reference be cited?
Reply: Accept. This indicator was not determined by the author themselves but was referenced from literature. Refer to the IPCC's classification method for possibilities (Sun et al., 2007) and other researchers' methods (Duan and Zhou, 2011); Wang et al., 2023). I will list these references in the article as references.
- - lines 188-189: repetition of line 137?
Reply:There is indeed a repetition from before, I deleted the duplicates in the results (lines 188-189).
- - pay attention to spaces between characters in lines 205, 209 and 220.
Reply: I added a space of 205 lines after 'between', 209 lines after “4.5”; 220 lines before “Analysis”.
- - Figure 8 cite before Figure 7 in the text; why?
Reply: I changed the position of Figures 7 and 8.
- - lines 203 and 244: "mali" not in italics.
Reply:Accept. Made "mali" in italics.
- - line 254: from largest to smallest values?
Reply:The maximum and minimum area of a highly suitable region would be 1,457,640 km2 and 1,216,426 km2, respectively in the scenarios of 2050s RCP2.6 and 2070s RCP8.5.
- - line 322: ...temperatures are the... factors... that affect.
Reply: Accept. Change the sentence to “So, the temperatures in November, March and April are the important factors that affect the beginning and end of overwintering.”
- the reference of Lu et al. 2022 (Bulletin of Entomological Research) could be cited in the discussion.
Reply: Accept. Thanks for you remind. I download this literature, which provides important insights for our article. I added it into the first paragraph in discussion part. Lu's prediction of Agrilus mali in China is consistent with our prediction.
References
- Ma ZL, 2020. Flight spread capability and attractant application technology research of Agrilus mali Matsumura. Xinjiang Agriculture university.(In Chinese)
- Cui XN, Yi ZH, Wang M, Liu DG, Liao SJ, Xu Z, 2016. Maturation Feeding Preference of Adult Agrilus mali and Related Host Plant Leaf Volatiles. Scientia Silvae Sinicae, 11(52): 96-106, (In Chinese)
- Li ML, Zhang ZQ, Chang Y. 2017. Addendum of biological characters of adult Agrilus mali Matsumura and rearing of larve. 32(6):214-219(In Chinese)
- Wang, R., Xia, Y., Shen, Z., Wang, Y., Zhou, X., Xiang, M., Yang, Y., 2023. Genetic diversity analysis and potential suitable habitat of Chuanminshen violaceum for climate change. Ecological Informatics 77, 102209.
- Duan J Q,Zhou G S, 2011.Potential distribution of rice in china and its climate characteristics. Acta Ecologica Sinica,31( 22) : 6659-6668.(In Chinese)
- Sun Y, Wang CK, Xu HM, Yu L, Luo Y, 2007. Facing the Climate Challenge: Interpretation of the Fourth Assessment Report of the Intergovernmental Panel on Climate Change (IPCC). China Journal of Disaster Reduction, (7):8-9. (In Chinese)
- Lu Z, Liu X, Wang T, Zhang P, Wang Z, Zhang Y, Kriticos DJ, Zalucki MP. Malice at the Gates of Eden: current and future distribution of Agrilus mali threatening wild and domestic apples. Bull Entomol Res. 2022 Dec;112(6):745-757. doi: 10.1017/S000748532200013X. Epub 2022 Apr 13. PMID: 35414375.
Reviewer 2 Report
Comments and Suggestions for Authors
Dear Authors, I have comprehensively reviewed the manuscript entitled: “Potential ecological distribution of the beetle Agrilus mali Matsumura (Coleoptera: Buprestidae) in China underthree climate change scenarios, with consequences for commercial and wild apple forests”. The manuscript provides an innovative perspective on how climate change may affect the distribution of invasive pests in commercial agricultural crops.
Specific recommendations:
Introduction
Overall, this section has a good structure, I think it is very well written. Nevertheless, I would recommend including a final paragraph where you specifically detail 2 things: 1) potential beneficiaries of the information generated in the research and 2) what SDGs your research relates to.
Line 100: A. mali. Scientific names should always be in italics.
Materials and Methods
Historic and current locations of A. mali
In this part of the methodology, the data cleaning protocol is not specified. It is not mentioned whether duplicates were removed, the precision of the coordinates used (digits after the comma) is not detailed, it is not mentioned whether spatial filtering was performed using a minimum distance or considering one sampling point per pixel. There is no mention of the percentage of data used for calibration and evaluation.
Please, these processes are of utmost importance to be detailed as they have a great influence on the quality of the generated models.
Environmental variables
The authors mention that they used the variables listed in table 1, but the method of variable selection is not clearly stated.
On the other hand, among the variables they include bio8, bio9, bio18, bio19; these variables are scientifically known to generate noise and problems in spatial predictions. Therefore, it is not advisable nor would I want to include them before the selection process.
Finally, and most importantly, I am very concerned about the resolution used. WorldClim variables are currently available at 1km resolution, which considering the minimum number of presence data used is the best resolution to use. Additionally, the geographic scale is manageable considering that level of resolution.
I suggest re-running the models at this resolution to ensure that the resulting maps are of the quality needed for decision making.
Modeling methods and statistical analysis
The authors used the Maxima entropy (Maxent) algorithm, but do not specify technically why they chose it. It is important to mention that scientifically it is not recommended to use this algorithm with the default settings because it has been found to generate serious problems of overestimation and extrapolation of predicted areas.
Currently, there are many modelling approaches that include testing different configurations with Maxent, for example, I suggest that the authors use multiple configurations so I recommend using the Kuenm package (https://github.com/marlonecobos/kuenm)
On the other hand, I could see that as an evaluation metric they used AUC. This metric is not advisable to use when there are no true absences, so it is preferable to use more metrics among which I ask the authors to use at least two additional metrics among them could be: partial AUC, Kappa, AICc, TSS.
Calculation of geometric center and displacement
The formulas used should not be copied as images, they should be placed using the formula option in WORD, as they need to be of the best possible quality.
Results
This section does not have a good structure, both the graphs and tables do not have the right quality and order. In addition, I do not recommend doing the analysis by provinces, because it is extensive and ineffective.
As a reference, I recommend analysing the presentation of the following pest-based documents:
https://peerj.com/articles/10690/
https://link.springer.com/article/10.1007/s13744-020-00840-4
https://esajournals.onlinelibrary.wiley.com/doi/full/10.1002/ecs2.3714
Author Response
Dear reviewer,
We are grateful very much to your comments for the manuscript. You have provided many suggestions that are beneficial for improving the manuscript. These suggestions can help us avoid some shortcomings. According with your advice, we amended the relevant pat in manuscript. Some of your questions were answered below.
- Introduction
Overall, this section has a good structure, I think it is very well written. Nevertheless, I would recommend including a final paragraph where you specifically detail 2 things: 1) potential beneficiaries of the information generated in the research and 2) what SDGs your research relates to.
Accept, we have added the following content to the last paragraph of the introduction: “In China, a large number of apple growers, planting enterprises, and seedling cultivation enterprises may conduct early monitoring based on our data. Once discovered, measures can be taken as soon as possible to avoid causing greater losses. The yield of apples taken from pests can provide people with more food and nutrition sources, which is in line with the second goal of the United Nations Sustainable Development Goal - eradicating hunger.”
- Line 100: mali. Scientific names should always be in italics.
Accept. Modified
- Materials and Methods
Historic and current locations of A. mali
In this part of the methodology, the data cleaning protocol is not specified. It is not mentioned whether duplicates were removed, the precision of the coordinates used (digits after the comma) is not detailed, it is not mentioned whether spatial filtering was performed using a minimum distance or considering one sampling point per pixel. There is no mention of the percentage of data used for calibration and evaluation.
Please, these processes are of utmost importance to be detailed as they have a great influence on the quality of the generated models.
Accept. We have made modifications to 2.1 Historic and current locations of A. mali in 2. Materials and Methods, detailing the basis for data filtering. As follows: Accordingly, 194 locations where A. mali was present or recorded were obtained. Repetitive, fuzzy, and adjacent records were deleted as per the requirements of MaxEnt. ENMTOOLs was used to calculate the distance between each grid center (30 arc-second) and each distribution data, then distribution data closest to the center in each grid was retained. All the retained 128 records were imported into Microsoft Excel software (Microsoft Office 2010) and saved in CSV format.
The accuracy of the latitude and longitude we have chosen is 2 decimal places.
- Environmental variables
The authors mention that they used the variables listed in table 1, but the method of variable selection is not clearly stated.
On the other hand, among the variables they include bio8, bio9, bio18, bio19; these variables are scientifically known to generate noise and problems in spatial predictions. Therefore, it is not advisable nor would I want to include them before the selection process.
Finally, and most importantly, I am very concerned about the resolution used. WorldClim variables are currently available at 1km resolution, which considering the minimum number of presence data used is the best resolution to use. Additionally, the geographic scale is manageable considering that level of resolution.
I suggest re-running the models at this resolution to ensure that the resulting maps are of the quality needed for decision making.
Accept. In our manuscript, the variables bio8, bio9, and bio18 were not selected. We have chosen the variable bio19. We referred to the Using Worthington's method for variable selection(39. Worthington, T.A.; Zhang, T.J.; Logue, D.R.; Mittelstet, A.R.; Brewer, S.K. Landscape and flow metrics affecting the distribution of a federally-threatened fish: Improving management, model fit, and model transferability. Ecological Modelling 2016, 342, 1-18, doi:10.1016/j.ecolmodel.2016.09.016.). Bio19 is precipitation of coldest quarter, whose change may affect extreme low temperatures. The impact of extreme low temperatures on insect colonization is enormous, so we did not abandon this climate factor.
On the issue that you are most concerned about, we have made changes based on your feedback. We have re-run the model at a resolution of 1Km to obtain a more accurate distribution range, resulting in changes to the figures and tables in the manuscript. The values have also changed.
The spatial resolution of the data was 30 arc seconds (1km).
- Modeling methods and statistical analysis
The authors used the Maxima entropy (Maxent) algorithm, but do not specify technically why they chose it. It is important to mention that scientifically it is not recommended to use this algorithm with the default settings because it has been found to generate serious problems of overestimation and extrapolation of predicted areas.
Accept. We have fully described the advantages and applicable conditions of the MaxEnt model in the introduction, please refer to the fourth paragraph of the introduction. Agrilus mali is a long-lived wood-boring pest, with one generation per year in most areas; At present, no one has been able to raise a complete generation under artificial conditions, so there is currently no data support for its developmental threshold temperature, effective accumulated temperature, etc. We have obtained all the distribution point information of it in China. Therefore, these have become the reason why we chose the MaxEnt model.
Currently, there are many modelling approaches that include testing different configurations with Maxent, for example, I suggest that the authors use multiple configurations so I recommend using the Kuenm package (https://github.com/marlonecobos/kuenm)
On the other hand, I could see that as an evaluation metric they used AUC. This metric is not advisable to use when there are no true absences, so it is preferable to use more metrics among which I ask the authors to use at least two additional metrics among them could be: partial AUC, Kappa, AICc, TSS.
Accept. According to expert opinions, Kuenm package has been selected; Comparison of different configuration results was conducted for two parameters RM and FC, and the most suitable parameter configuration for this model was selected based on AUC and AICc indicators. The Kuenm R package was used to optimize the regularization multiplier (RM) (0.5, 1, 1.5, 2, 2.5, 3, 3.5, 4) and feature combination (FC, including 30 types) of the model, and the optimal setting of the minimum information criterion AICc value (delta. AICc) among 240 results was selected.
- Calculation of geometric center and displacement
The formulas used should not be copied as images, they should be placed using the formula option in WORD, as they need to be of the best possible quality.
Accept. Modified.
- Results
This section does not have a good structure, both the graphs and tables do not have the right quality and order. In addition, I do not recommend doing the analysis by provinces, because it is extensive and ineffective.
As a reference, I recommend analysing the presentation of the following pest-based documents:
https://peerj.com/articles/10690/
https://link.springer.com/article/10.1007/s13744-020-00840-4
https://esajournals.onlinelibrary.wiley.com/doi/full/10.1002/ecs2.3714
Accept. We adjusted the order of the charts and tried to use high-quality images as much as possible. Thank you very much for providing excellent literature to assist me in improving the content of the results analysis section.
We deleted 6 AUC charts and 2 centroid maps.
Round 2
Reviewer 2 Report
Comments and Suggestions for Authors
Dear authors.
Thank you very much for addressing each of my observations for the improvement of your manuscript. I believe that the changes made have been thorough and ensure the scientific quality of the information you are providing to the community.
I recommend publishing the manuscript in its current state.
Best wishes.